# Loss of CFTR Reverses Senescence Hallmarks in SARS-CoV-2 Infected Bronchial Epithelial Cells

**DOI:** 10.3390/ijms25116185

**Published:** 2024-06-04

**Authors:** Flavia Merigo, Anna Lagni, Federico Boschi, Paolo Bernardi, Anita Conti, Roberto Plebani, Mario Romano, Claudio Sorio, Virginia Lotti, Andrea Sbarbati

**Affiliations:** 1Anatomy and Histology Section, Department of Neurosciences, Biomedicine and Movement Sciences, University of Verona, 37134 Verona, Italy; flavia.merigo@univr.it (F.M.); paolo.bernardi@univr.it (P.B.); anita.conti@univr.it (A.C.); andrea.sbarbati@univr.it (A.S.); 2Microbiology Section, Department of Diagnostic and Public Health, University of Verona, 37134 Verona, Italy; anna.lagni@univr.it; 3Department of Engineering for Innovation Medicine, University of Verona, 37134 Verona, Italy; federico.boschi@univr.it; 4Laboratory of Molecular Medicine, Center for Advanced Studies and Technology (CAST), Department of Medical, Oral and Biotechnological Sciences, “G. d’Annunzio” University of Chieti-Pescara, 66100 Chieti, Italy; roberto.plebani@unich.it (R.P.); mario.romano@unich.it (M.R.); 5General Pathology Section, Department of Medicine, University of Verona, 37134 Verona, Italy; claudio.sorio@univr.it

**Keywords:** autophagy, CFTR, cystic fibrosis, lipid, lipolysosome, SARS-CoV-2, senescence, ultrastructure

## Abstract

SARS-CoV-2 infection has been recently shown to induce cellular senescence in vivo. A senescence-like phenotype has been reported in cystic fibrosis (CF) cellular models. Since the previously published data highlighted a low impact of SARS-CoV-2 on CFTR-defective cells, here we aimed to investigate the senescence hallmarks in SARS-CoV-2 infection in the context of a loss of CFTR expression/function. We infected WT and CFTR KO 16HBE14o-cells with SARS-CoV-2 and analyzed both the p21 and Ki67 expression using immunohistochemistry and viral and p21 gene expression using real-time PCR. Prior to SARS-CoV-2 infection, CFTR KO cells displayed a higher p21 and lower Ki67 expression than WT cells. We detected lipid accumulation in CFTR KO cells, identified as lipolysosomes and residual bodies at the subcellular/ultrastructure level. After SARS-CoV-2 infection, the situation reversed, with low p21 and high Ki67 expression, as well as reduced viral gene expression in CFTR KO cells. Thus, the activation of cellular senescence pathways in CFTR-defective cells was reversed by SARS-CoV-2 infection while they were activated in CFTR WT cells. These data uncover a different response of CF and non-CF bronchial epithelial cell models to SARS-CoV-2 infection and contribute to uncovering the molecular mechanisms behind the reduced clinical impact of COVID-19 in CF patients.

## 1. Introduction

Virus-induced senescence (VIS) may be regarded as an antiviral defense mechanism, or a means to promote virus replication [1,2]. Senescent cells are mainly characterized by beta-galactosidase (B-gal) activity, cyclin kinase inhibitor protein (e.g., p16INK4a and p21Cip1) activation, and resistance to apoptosis [3]. In addition, lipofuscin content has been considered as a senescence marker, defined as lipofuscin granules using light microscopy, or residual bodies using transmission electron microscopy [4]. Moreover, senescent cells contribute to chronic inflammation, releasing pro-inflammatory chemokines, cytokines, metalloproteases, reactive oxygen species (ROS), metabolites, miRNAs, and the so-called senescence-associated secretory phenotype (SASP) [5,6,7]. Interestingly, SARS-CoV-2 infection has recently been shown to induce cellular senescence in vivo [8,9,10,11] and has been reported to alter the different host pathways [12]. Moreover, the direct induction of biomarkers of senescence through infection with SARS-CoV-2 was demonstrated in vitro in different cell types [13,14]. SARS-CoV-2 is indeed responsible for senescence induction and exacerbation of the SASP, and cellular senescence has been proposed as a critical regulator of SARS-CoV-2-evoked hyperinflammation [13,15]. The importance of senescence in SARS-CoV-2 infection is also highlighted in several clinical trials on senolytics, with some encouraging initial results [8,16,17].

Cystic fibrosis (CF) is the most common life-shortening genetic recessive disease in Caucasian people, resulting from a mutation in a gene located on chromosome 7 (7q31.2) that encodes a protein termed Cystic Fibrosis Transmembrane Conductance Regulator (CFTR) [18]. The CFTR protein is widespread in the epithelial cell apical membrane (lungs, upper respiratory tract, pancreas, liver, gallbladder, intestines, sweat glands, and reproductive tract), exerting a key role in chloride (Cl^−^) and bicarbonate (HCO_3_^−^) transport, which are essential for the osmotic balance of the mucus and its viscosity [19,20,21]. The CFTR protein is also involved in the regulation of other ion channels, specifically the epithelial sodium (Na^+^) channel (ENaC), which is essential for salt absorption [22]. People with cystic fibrosis (pwCF) who are recurrently affected by bacterial infection and neutrophilic inflammation in their airways [23,24] have been proposed to exhibit a senescence-like phenotype [25,26]. Fisher et al. [27] reported the induction of the arrest or quiescence of the epithelial cell cycle in pwCF airways due to the presence of neutrophil elastase (NE), which can also result in cellular apoptosis or senescence [28,29]. The increased expression of senescence markers, such as p16 (*CDKN2A*), γ-H2AX, and phospho-checkpoint kinase 2 (CHEK2), was reported in pwCF primary bronchial epithelial cells [25] and, in addition, neutrophils derived from pwCF were reported to express p21 (*CDKN1A*) [30]. Despite consistent data supporting the occurrence of cellular senescence in CF [26], the relationship between CFTR loss-of-function and related downstream signaling with the transition towards senescence is scarcely known.

Since the previously published data [31,32] highlighted a low impact of SARS-CoV-2 on CFTR-modified cells, our objectives were to analyze whether CFTR KO 16HBE14o-cells show a senescence-like phenotype and to uncover a possible interaction between senescence and CFTR expression/function in SARS-CoV-2 infection, using histochemical, molecular, and morphological analyses.

## 2. Results

Here, we used histochemical, immunohistochemical, and molecular biology approaches to demonstrate the in vitro expression of cellular senescence biomarkers in mock and SARS-CoV-2-infected WT and CFTR KO 16HBE14o-cells. We also evaluated the morphological changes occurring in these cells at the ultrastructural level upon SARS-CoV-2 infection.

### 2.1. High Accumulation of Oil Red O Staining in CFTR KO 16HBE14o-Cells

Since lipids have been recently recognized as key players in cellular senescence [33], we first investigated the presence of lipids in WT and CFTR KO 16HBE14o-cells at different culture times using Oil Red O (ORO) staining. The perinuclear and cytoplasmic aggregates, stained by ORO as red granules, were clearly visible in both cell lines. However, the ORO staining appeared different in the two cell lines and varied over time (Figure 1A). A quantitative analysis of the ORO-covered area revealed higher values in CFTR KO cells than in WT cells at all analyzed time points, which increased significantly at 1, 6, 48, and 72 h, supporting more severe lipid accumulation in CFTR KO cells (Figure 1B).

### 2.2. Expression of the p21 Senescence Marker in Mock- and SARS-CoV-2-Infected WT and CFTR KO 16HBE14o-Cells

Since dysfunctional lysosomes containing lipofuscins are reported to continuously accumulate in senescent cells that cannot proliferate, we examined using immunohistochemistry the cellular expression of p21, a well-known nuclear senescence biomarker.

As expected, the p21 immunoreactivity was concentrated exclusively in the nucleus with a variable intensity staining that could be marked and diffuse throughout the nucleus, except for the nucleoli, or mild and nonhomogeneous diffuse but limited to only a few areas. p21-immunoreactive (IR) cells were observed in WT and CFTR KO 16HBE14o-cells before and after SARS-CoV-2 infection (Figure 2A). In mock-infected samples, p21-IR cell numbers were significantly more abundant in the CFTR KO clone than in WT cells at all time points, suggesting a more prominent senescent pattern was associated with CFTR-loss-of-function (Figure 3A). At 24 h, immunoreactivity to p21 was observed in 51% and 32% of CFTR KO and WT cells, respectively (Appendix A). A remarkable reversal of the situation was found after SARS-CoV-2 infection, where the percentage of p21-IR cells was higher in WT cells than in CFTR KO cells at all time points, with a statistically significant increase at 24 h post infection (hpi) (Figure 3B and Appendix A). For both WT and CFTR KO cells, we subtracted the percentage of p21-positive cells in mock infection conditions from the percentage detected following SARS-CoV-2 infection. In particular, SARS-CoV-2 infection led to an increasing trend of p21-IR cells in WT cells at all time points analyzed, while a decreasing trend of p21-IR cells was observed in CFTR KO cells at all time points analyzed (except at 3 h), suggesting a different impact of SARS-CoV-2 infection in the two cell lines (Figure 3C).

Analyzing p21 gene expression, mock-infected CFTR KO 16HBE14o-cells displayed a higher p21 mRNA expression than WT cells over time, reaching a statistically significant difference at 24, 48, and 72 h (Figure 4A). As for p21-IR, after SARS-CoV-2 infection, we found an inversion of the trend, with a significantly higher p21 gene expression in WT than in CFTR KO cells at 24, 48, and 72 h (Figure 4B). Analyzing the SARS-CoV-2 solo impact on WT cells, we observed an increase in p21 gene expression over time, in agreement with the recent literature reporting an induction of the senescence pathway after infection [8,14,15,34]. Interestingly, CFTR KO cells had SARS-CoV-2 downregulated p21 gene expression at all time points analyzed, suggesting a peculiar and opposite effect of the virus on the senescence pathway of CFTR-modified cells. We measured the statistically significant differences by comparing the results of the SARS-CoV-2 impact in WT cells and CFTR KO cells. (Figure 4C).

### 2.3. Expression of the Ki67 Proliferation Marker in Mock- and SARS-CoV-2-Infected WT and CFTR KO 16HBE14o-Cells

The Ki67 antigen (MKI67 gene) is a marker of cellular proliferation that is downregulated in senescent cells. Anti-Ki67 antibody immunostaining permitted us to identify a nuclear diffuse, intensively, or moderately marked staining (Figure 2B). In mock-infected cells, the percentage of immunoreactive (Ki67-IR^+^) cells was higher in WT cells than in the CFTR KO cell population from 24 to 72 h of culture (Figure 5A) indicating, in agreement with the p21-IR data, a high number of non-proliferative cells in CFTR-KO cells. At 3 h, the percentage of positive cells was comparable in the two cell lines. The difference at 48 and 72 h was statistically significant. In SARS-CoV-2-infected cells, the percentage of Ki67-IR^+^ cells was higher in CFTR KO cells than in WT cells at all time points, except for 72 h, where the opposite occurred (Figure 5B and Appendix A). The exposure to SARS-CoV-2 exerted different effects on the proliferation of the two 16HBE14o-cell lines analyzed. Thus, when we isolated the impact of the virus on the cell proliferative capacity by subtracting the percentage of Ki67^+^ mock- from SARS-CoV-2-infected cells, we identified an upward trend over time for CFTR KO cells. Conversely, WT cells revealed an opposite, decreasing trend (Figure 5C), suggesting a stimulatory effect of the virus on cell proliferation in cells lacking CFTR.

### 2.4. Morphological Characteristics of WT and KO 16HBE14o-Cells

At an early time of culture, mock-infected WT cells showed normal morphological characteristics: abundant and dense cytoplasm that contained short strips of endoplasmic reticulum (ER), small and normal shaped mitochondria, a well-developed Golgi apparatus, and a few clear vacuoles (Figure 6A,B). Rare lipid droplets, individually localized and associated with granular, electron-dense deposits forming structures identifiable as residual bodies and multivesicular bodies (MVBs), were visible in the cytoplasm (Figure 6C). In addition, some cells showed cytoplasmic, irregular-shaped, autophagosomes with a content of membranous matter or cytoplasmic material (Figure 6D,E).

In comparison, mock-infected CFTR KO cells showed moderate dilatations in the ER and large deposits of electron-dense granules (Figure 6F). Moreover, lipid droplets were individually localized in proximity to or in contact with the mitochondria or aggregated at the base of the nucleus. These structures were variable in size and were frequently surrounded by an external electron-dense membrane, a feature typical of lipolysosomes (Figure 6G). In the cytoplasm, granular electron-dense residual bodies were present (Figure 6H), and some cells also contained onionskin-like structures (Figure 6I), surrounded by electron-dense granules (Figure 6J).

At a late time of culture, mock-infected WT cells presented cytoplasmic vacuoles with a prominent limiting membrane, a rounded and clear shape, and a relatively low granular content (Figure 7A,B). Electron-dense aggregates and residual bodies varied in size and were often close to the mitochondria (Figure 7C). Autophagosomes were still present in the cytoplasm with the same shape described at an early stage of culture (Figure 7D,E).

In CFTR KO mock-infected cells, the number and size of residual bodies were increasingly pronounced compared with those at the early hours of culture (Figure 7F). Lipolysosomes surrounded by several concentric bilayers of membranes (multilamellar profile), which are always located near the nucleus, were observed (Figure 7G). A few large autophagosomes were visible with the heterogeneous content (Figure 7H), ranging from electron-opaque amorphous deposits to vesicles with cytoplasmic content, multilamellar membranes, or structures with lipid content (Figure 7I,J).

The main differences between an early and late time of culture are summarized in Table 1.

### 2.5. Morphological Modifications of WT and KO 16HBE14o-Cells in Response to SARS-CoV-2 Infection

At the early stages of SARS-CoV-2 infection, CFTR WT cells maintained a physiological morphology with a dense matrix cytoplasm and a well-developed electron-dense ER, due to the numerous polyribosomes attached to its surface. Numerous electron-dense granules were scattered in the cytoplasm or aggregated to form small deposits (Figure 8A). Double-membrane vesicles (DMVs), present in different sizes in the cytoplasm, with an electron-dense granular matrix and sometimes with a spacing between their membranes, were considered as viral replicative structures (Figure 8B,C). Small-sized residual bodies were observed in some cells throughout the cytoplasm and in proximity to the nucleus (Figure 8D). Autophagosomes were found in a few cells, filled with various cytoplasmic contents and vesicles with a dense matrix, probably corresponding to the lipid droplets or lysosomes (Figure 8E,F).

Similarly to SARS-CoV-2-infected CFTR WT cells, CFTR KO cells showed a moderately well-developed ER, rich in ribosomes on the outer surface and extensive dilatations increasing with the time of infection (Figure 8G). Small and irregularly shaped mitochondria with swollen cristae, numerous small vesicles, and diffuse or accumulated electron-dense granules were observed (Figure 8H). Membrane-bound organelles that were lysosome-like and MVBs were identified in the cytoplasm of some cells (Figure 8I) and lipid droplets were found only occasionally (Figure 8J). In contrast with SARS-CoV-2-infected WT cells, many CFTR KO cells displayed extensive autophagic vacuoles with a content of structureless masses composed of different materials (Figure 8K,L).

Late in the infection, CFTR WT cells showed single-membrane, matrix-free vesicles containing dense particles, identified according to their size and appearance as virion-containing vesicles (Figure 9A). MVBs and lysosomes were frequently observed in the cytoplasm (Figure 9B). From 24 hpi, lipid droplets were scattered in the cytoplasm, which at 72 hpi aggregated near or in continuity with the electron-dense deposits (Figure 9C,D). Large autophagosomes containing densely packed vesicles resembling virus replication structures increased from 48 hpi onwards (Figure 9E,F).

The overall characteristics of the infected CFTR KO cells did not differ with increasing infection time (Figure 9G), except for the presence of DMVs in the cytoplasm, which, in continuity with the ER membranes, had electron-lucent areas or contained circular membranes (Figure 9H,I). Nonetheless, electron-lucid vesicles containing mature virus particles were never identified. The size of lipid droplets increased with infection time and tended to be isolated (Figure 9J). Large autophagosomes were observed, with a wide variety of contents: matrix of cytoplasmic origin, vesicles with probable mitochondrial origin, or multilamellar structures (Figure 9K,L).

The main differences between an early and late time of culture are summarized in Table 2.

### 2.6. Characterization of SARS-CoV-2 Stage of Infection in WT and CFTR KO 16HBE14o-Cells

To associate the morphological data with the actual virion replication, we assessed the SARS-CoV-2 viral content in both the supernatant and cell lysate of WT and CFTR KO 16HBE14o-cells. The SARS-CoV-2 viral load followed a time-dependent trend, with lower expression in CFTR KO than in WT cells in all the time points, as reported by multiplex real-time RT–PCR of supernatants, showing a statistically significant difference at both 48 and 72 hpi (Figure 10A). Consistent with this, ORF1ab intracellular mRNA levels were significantly reduced at 72 hpi (Figure 10B). Interestingly, the presence of viral particles in the supernatant was correlated with the intracellular mRNA expression of the ORF1ab viral gene suggesting that both cell types are competent to permit the viral cycle, which nevertheless occurred at a significantly lower efficiency in CFTR KO cells (~20-fold reduction in virus production in comparison with WT 16HBE14o-cells) (Appendix A). Given the reduced SARS-CoV-2 infection efficacy observed in KO compared to CFTR WT cells, we tried to dissect the steps of the viral infection likely to be affected by the ablation of CFTR expression. The reduced viral production might derive from either reduced entry or less effective assembly/maturation of virions within the cell or a combination of both mechanisms. ACE2 is one of the major entry receptors for SARS-CoV-2, thus we performed a flow cytometry analysis of its constitutive membrane expression which revealed a higher ACE2 protein expression in WT compared with CFTR KO cells (Appendix A).

We further investigated the expression of the SARS-CoV-2 Nucleocapsid protein (N), a marker of new viral particle production, in WT and CFTR KO 16HBE14o-cells using immunohistochemistry. Both SARS-CoV-2-infected cell lines displayed a similar pattern of N labelling that was primarily observed in the cytoplasm but was also evident in the nucleus. No positivity was observed in the mock-infected cells or 1 h post infection (Appendix A). N protein-expressing cells were present in both cell lines 24 h post-SARS-CoV-2-infection onwards, although the positivity was more pronounced in WT than in CFTR-KO cells at all time points analyzed (Appendix A), in line with the reduced viral load measured in the extracellular milieu.

## 3. Discussion

The expression of CFTR in diverse cell types and organs is widely recognized, with research indicating multiple functions beyond its role as an anion exchange channel. In fact, being expressed by several cell types as epithelial cells [35,36,37,38], endothelial cells [39,40,41] fibroblasts [42,43], and leukocytes [44,45], it is involved in multiple functions. For instance, CFTR plays a crucial role in the transport of bicarbonate, contributing to the regulation of the airway surface liquid pH [22]. Moreover, CFTR has been implicated in cellular processes, such as apoptosis [46], carcinogenesis [47], and inflammation [48]. Research indicates its involvement in host defense against viral infections, such as SARS-CoV-2, where CFTR deficiency correlates with decreased susceptibility [31,49,50,51]. In this study, we compared senescence markers in WT and CFTR-knockout (KO) human bronchial 16HBE14o-cells. Initially, using the ORO staining, we evaluated the presence of lipid droplets known to be involved in the initiation and maintenance of senescence [33], and in the formation of lipofuscin granules, which are considered a senescence marker accumulating inside the lysosomes of non-replicative cells [52,53,54]. We observed a significantly higher percentage of ORO-stained areas in CFTR KO than in WT cells at all time points analyzed, suggesting a significant morphological alteration in the CFTR-modified cell line. An additional biomarker (p21) supports the concept of a senescence-like phenotype of CFTR-modified cells. We further assessed cellular proliferation by Ki67 immunostaining, recording a higher positivity in WT than in CFTR KO cells, suggesting a reduced proliferative drive in KO cells. These findings support the previously reported increase in senescence markers expression in CF cells [25,55].

Autophagy is a homeostatic mechanism of lysosome-based degradation linked to cellular senescence. Thus, impaired autophagy may be an important cause for increasing senescence in CFTR KO cells. Using a TEM analysis, we observed a different ultrastructural morphology between CFTR KO and WT cells: lipid droplets were consistently observed in CFTR KO cells but rarely in WT cells, confirming the results obtained using ORO staining. Most lipid droplets identified in CFTR KO cells were lipolysosomes, externally surrounded by an electron-dense membrane. An increased number of these structures was associated with impaired lipid degradation due to lysosomal dysfunction [56,57]. Incomplete lysosomal degradation leads to the accumulation of lipofuscin granules [53], which have been recently reported not to be an inert product of the cell, but rather to alter the cell metabolism at multiple levels, inhibiting the proteasome, hindering autophagy and lysosomal degradation, and acting as a metal ion reservoir, resulting in ROS generation [58] and apoptotic cell death [59]. Based on these observations, it is possible to hypothesize that the presence of lipolysosomes and lipofuscin granule accumulation in CFTR KO may be strictly related to senescence and autophagy [60], probably due to the altered lysosomal activity. In this regard, senescence has been shown to reduce the autophagic flux [61]. Notably, CF cell lines and CFTR knockout animal studies indicate that the loss of CFTR is sufficient to generate a proinflammatory environment [62], which can lead to the ROS-mediated activation of transglutaminase-2 (TGM-2) and the inactivation of the Beclin-1 complex, resulting in autophagy impairment [55,63,64]. Since the results in mock-infected cells suggested the activation of a senescence-like pathway in CFTR-modified cells, and previously published data highlighted a low impact of SARS-CoV-2 on these cells [31,32,49,50,65], we aimed to analyze a possible interaction between senescence pathways, CFTR expression/function, and SARS-CoV-2 infection. Viral infections can trigger premature cellular senescence, known as virus-induced senescence (VIS). It was reported that some viruses, such as human immunodeficiency virus, measles virus, respiratory syncytial virus, and influenza virus, can take advantage of senescence as a mechanism of spread in infected organisms [66], triggering senescence to increase their replication rate [67]. Recently, it was demonstrated that SARS-CoV-2 also evokes cellular senescence as a primary stress response in infected cells, activating the DNA damage response and exacerbating SASP-related paracrine senescence [8,9,13,15]. Consistent with these findings, the present study demonstrated in vitro the induction of senescence biomarkers by SARS-CoV-2 infection in non-CF cells: the combined evaluation of p21 and Ki67 antigens revealed that SARS-CoV-2 increased p21 and, concomitantly, reduced the percentage of proliferative cells over time, in accordance with the reported evidence of SARS-CoV-2-induced senescence [13,15]. Conversely, SARS-CoV-2-infected CFTR KO cells showed decreased p21 immunoreactivity and gene expression, combined with an increased Ki67 immunoreactivity that was at variance with that recorded in the isogenic cells expressing native CFTR, indicating an opposite viral impact on the senescence pathway.

Moreover, we observed that both the viral load, measured in the supernatant, and the ORF1a,b gene expression, evaluated intracellularly, were lower in KO cells than in CFTR WT cells, confirming the results reported in our previous work [32]. Consistent with these results, a flow cytometry analysis of ACE2, a SARS-CoV-2 entry receptor, revealed a significantly lower constitutive membrane expression in KO cells than in CFTR WT cells, suggesting lower susceptibility. Nevertheless, the IHC results showed a reactivity to the N SARS-CoV-2 protein antibody in both WT cells and CFTR KO cells at all time points of the infection tested, although the reactivity was more pronounced in WT cells than in KO cells. These results indicate that the ablation of CFTR expression can affect different steps involved in the viral cycle, from viral entry to replication. Moreover, its opposite impact on senescence patterns may suggest the involvement of cellular senescence in the virus replication cycle.

Indeed, at the ultrastructural level, when comparing SARS-CoV-2- and mock-infected WT cells at the late times of infection, we found a greater presence of lipid droplets and autophagosomes containing viral replicative structures within the internal matrix. These findings may represent the cellular changes induced by the virus to facilitate its replication, as demonstrated in recent in vitro studies [68,69]. The virus may exploit endogenous lipid components in various forms (i.e., lipoproteins and exosomes) as “Trojan horses” to facilitate immune evasion in their systemic dissemination [70,71]. In contrast, SARS-CoV-2-infected CFTR KO cells exhibited fewer lipid droplets scattered in the cytoplasm than their mock-infected counterparts, which were not identifiable as lipolysosomes due to the absence of an external membrane. Furthermore, we observed viral replicative structures, called DMVs, in CFTR KO cells at a later time than in WT cells and without mature forms of virus, usually contained in clear, matrix-free vesicles, as observed in WT cells. These results and the measures of intracellular and extracellular viral mRNA suggest that CFTR KO cells display an impaired viral replication cycle and require longer times to complete all stages of the viral infection cycle.

Interestingly, the removal of senescent cells reduced the viral load and attenuated pulmonary and systemic inflammation in SARS-CoV-2 infection [72]. Thus, senotherapeutics, drugs targeting senescent cells [73], seem to play an important role in fighting viral infection. In particular, senolytics (such as quercetin, fisetin, navitoclax, and dasatinib), acting by removing senescent cells, and senomorphic agents (such as rapamycin, metformin, anakinra, and tocilizumab), inhibiting some component of the SASP, mitigate age-related viral infection [66]. The CFTR phenotype appears to have a senotherapeutic impact on SARS-CoV-2 infection, mimicking the effect of the previously mentioned compounds and contributing to explaining the significantly different response of non-CF and CF cells to SARS-CoV-2 infection.

In conclusion, although further studies are required, the present work provides novel insights into the regulatory mechanisms of cellular responses to SARS-CoV-2 infection, which appear to involve the CFTR protein.

## 4. Materials and Methods

### 4.1. Cell Lines and Virus Strain

The human bronchial epithelial cell line 16HBE14o- was employed as a wild-type cell line, whereas a clone genetically engineered using CRISPR–Cas9 technology with complete deletion of the CFTR gene (knockout; KO) was used to simulate the CFTR nonsense mutations [74,75]. CRISPR/Cas9 ablation of CFTR expression was carried out by targeting the exon 1 of the CFTR gene in the human airway cell line 16HBE14o-. These cell lines were originally described by Plebani and colleagues [75]. The cells were cultured in MEM (Gibco, Thermo Fischer Scientific, Waltham, MA, USA) supplemented with 10% FBS (Euroclone, Milan, Italy) and 1% glutamine (Gibco, Thermo Fischer Scientific) and grown to a monolayer after passaging. We confirmed the successful CFTR knockout by evaluating the CFTR protein expression using a western blot analysis of both WT and CFTR KO 16HBE14o-cells (Appendix A).

The SARS-CoV-2 B.1 strain (hCoV-19/Italy/BO-VB12/2020|EPI_ISL_16978127) was isolated from a respiratory secretion from a COVID-19-positive adult male at S. Orsola Hospital (Bologna, Italy) in March 2020 and replicated as described by Ogando and colleagues [76].

### 4.2. Cell Culture and Infection

WT and CFTR KO 16HBE14o-cells were cultured in adhesion to reach approximately 70% confluence. Then, the cells were inoculated with SARS-CoV-2 at a multiplicity of infection (MOI) of 1 and incubated at 37 °C and 5% CO_2_ for 1 h. Next, the inoculum was removed, the cells were washed twice with PBS, and a fresh medium was added. After 3, 6, 24, 48, and 72 hpi, the media was discarded, and the cells were washed twice with PBS and harvested as needed for each technique. For the mock infection, media was added to the cell culture. Mock-infected cells were grown and harvested at the same time points as the infected cells.

### 4.3. Oil Red O Staining

We used Oil Red O (ORO) staining to evaluate the presence of lipids in mock-infected WT and CFTR KO 16HBE14o-cells. The cell layers were fixed with a Backer fixative (Bio-Optica, Milan, Italy) at 4 °C for 10 min, washed in running tap water, and dipped in Oil Red O solution (Bio-Optica, Milan, Italy) for 20 min. After washing in tap water, cells were counterstained with Mayer’s hematoxylin, washed in tap water, and mounted with an aqueous medium.

The technique was applied to specimens harvested at 1, 3, 6, 24, 48, and 72 h of culture. The quantification of the area covered by Oil Red O staining was performed on digital images acquired with the same objective (60×) of the optical Olympus BX51 microscope (Olympus, Tokyo, Japan). Ten digital images (dimensions: 1392 × 1040 pixels) were randomly selected in both WT and CFTR KO cells for each time point.

### 4.4. Immunohistochemistry

For the immunohistochemistry, mock- and SARS-CoV-2-infected CFTR KO and WT 16HBE14o-cells were cultured in adhesion on sterile rounded coverslips (13 mm) in a 24-well plate. After infection, the cells were washed twice with PBS and fixed for 2 h with formalin. To block nonspecific peroxidases, fixed cells were then incubated for 20 min in 3% H_2_O_2_ and after washing, they were incubated for 2 h in a blocking buffer (BSA 1%, Triton X100 0.3%; serum 2%). Then, the samples were incubated with anti-p21 (Abcam, Cambridge, UK, ab109520) or anti-Ki67 antibodies (Thermo Fisher Scientific, MA5-14520) or SARS Nucleocapsid Protein Antibody (Novus Biologicals, Centennial, CO, USA, NB100-56576). The ABC solution was then applied to the samples according to the Vectastain Elite ABC kit (Vector, Burlingame, CA, USA). The immunoreaction was developed using 3,3 diaminobenzidine (DAB) tetrahydrochloride (Dako, Milan, Italy), followed by hematoxylin counterstaining. Slides were mounted with the aqueous mounting media and then examined with an Olympus BX51 microscope (Olympus, Tokyo, Japan) equipped with a digital camera (DKY-F58 CCD, Yokohama, Japan). Immunohistochemistry performed without primary antibodies was considered as a negative control. The technique was applied to specimens harvested at 3, 24, 48, and 72 h of cell culture.

The quantification of cells that were immunoreactive for p21 and Ki67 was performed using manual counting of the positive nuclei in digital images acquired with the same objective (60×) of the optical Olympus BX51 microscope (Olympus, Tokyo, Japan). Ten digital images (dimensions: 1392 × 1040 pixels) were randomly selected in WT and CFTR KO cells both before and after SARS-CoV-2 infection for each time point.

### 4.5. Real-Time RT–PCR

The SARS-CoV-2 load in the supernatant was detected using the multiplex real-time PCR Allplex 2019-nCoV assay kit targeting the E, RdRp/S, and N genes (Seegene, Seoul, Republic of Korea) following the manufacturer’s instructions. Total RNA was extracted from cells with ReliaPrep™ RNA Miniprep Systems (Promega, Madison, WI, USA) and retrotranscribed into cDNA with iScript™ Reverse Transcription Supermix for RT–qPCR (Bio-Rad, Hercules, CA, USA). RT–qPCR was set up with PowerUp™ SYBR Green™ Master Mix (Applied BioSystems, Thermo Fisher Scientific) and then analyzed using real-time qPCR on a CFX96 Real-Time System (Bio-Rad) using primers targeting the SARS-CoV-2 ORF1ab gene (F: 5′-TGATGATACTCTCTGACGATGCTGT-3′; R: 5′-CTCAGTCCAACATTTTGCTTCAGA-3′) and senescence-associated CDKN1A (p21) mRNA (F: 5′-GACACCACTGGAGGGTGACT-3′; R: 5′-CAGGTCCACATGGTCTTCCT-3′). The relative gene expression was calculated using the ΔCT method and normalized to the expression of the housekeeping control genes. The −ΔΔCt method was used for absolute quantification. The technique was applied to the specimens harvested at 3, 24, 48, and 72 h.

### 4.6. Transmission Electron Microscopy (TEM)

For the ultrastructural examination, cell pellets of mock- and SARS-CoV-2-infected WT and CFTR KO 16HBE14o-cells were harvested and fixed for 1 h in 2% glutaraldehyde in a 0.1 M phosphate buffer (PB) and, after washing, postfixed for 1 h in 1% OsO4 diluted in 0.2 M K_3_Fe(CN)_6_. After rinsing in 0.1 M PB, the samples were dehydrated in graded concentrations of acetone and embedded in a mixture of Epon and Araldite (Electron Microscopic Sciences, Fort Washington, PA, USA). Ultrathin sections were cut at a 70 nm thickness on an Ultracut-E ultramicrotome (Reichert-Jung, Heidelberg, Germany), contrasted with lead citrate and observed on a Philips Morgagni 268 D electron microscope (Fei Company, Eindhoven, NL, USA) equipped with a Mega View II camera (Olympus, Tokyo, Japan) for the acquisition of digital images. The technique was applied to specimens harvested at 1, 3, and 6 (considered grouped as the early stages) and 24, 48, and 72 h of cell culture (considered grouped as the late stages).

### 4.7. Statistical Analysis

To evaluate the percentage of the ORO-stained area, a routine was written in MATLAB 9.5 (R2018b, Mathworks, Natick, MA, USA), considering positive pixels with a signal intensity above a visually determined threshold. The Kolmogorov–Smirnov test revealed that the data were not normally distributed; therefore, the differences between the WT and CFTR KO groups were analyzed using Wilcoxon’s test using MATLAB.

To analyze the difference between the number of p21- and Ki67-positive cells in mock-infected and SARS-CoV-2-infected conditions, Fisher’s exact test was performed using MATLAB.

The real-time PCR data were statistically analyzed using GraphPad Prism version 5 (GraphPad Software Inc., La Jolla, CA, USA). The comparisons were performed using 2-way ANOVA.

Differences were considered statistically significant when *p* < 0.05.

## 5. Conclusions

In summary, the present study is the first to demonstrate in vitro that CFTR KO 16HBE14o-cells exhibit an increased expression of senescence-associated biomarkers when compared with their WT counterparts at both the cellular and molecular levels. We also observed for the first time that SARS-CoV-2 infection, while increasing the expression of senescence markers in non-CF cells, has the opposite effect on CFTR KO cells. These results could be useful to better understand how altered CFTR expression/function affects SARS-CoV-2 replication and the mechanism(s) behind its reduced pathogenicity recorded in pwCF.

## Figures and Tables

**Figure 1 ijms-25-06185-f001:**
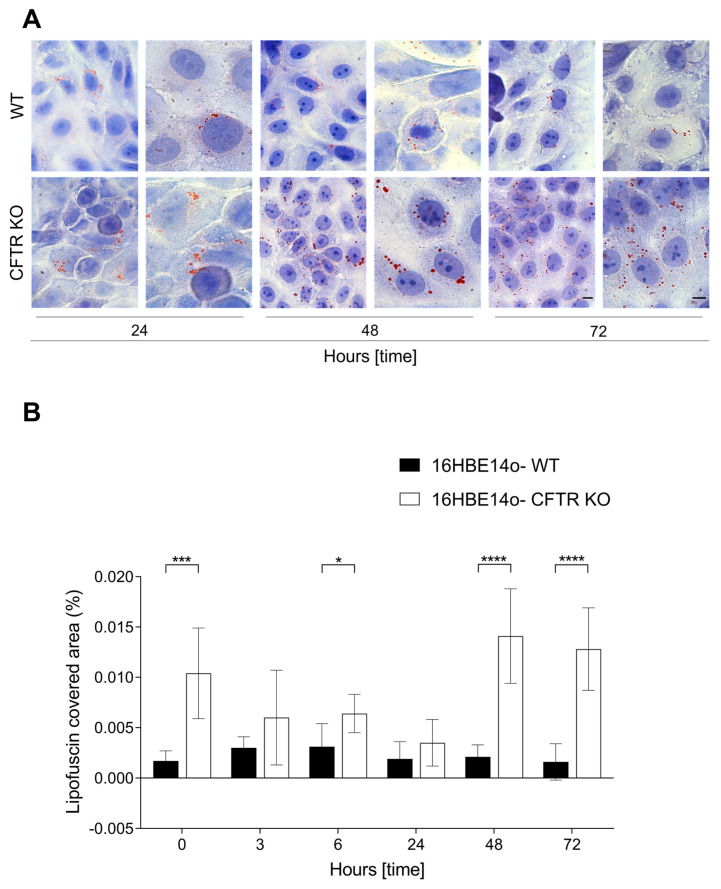
Oil Red O staining of mock-infected WT and CFTR KO 16HBE14o-cells. (**A**) Oil Red O-positivity in cells counterstained with Mayer’s hematoxylin was visible as the red dots scattered in the cytoplasm or in proximity to the nucleus. Representative images of both WT and CFTR KO cells at 24, 48, and 72 h are reported. Bars: 10 µm. (**B**) Quantitative analysis of the percentage of the area covered by Oil Red O. Data are presented as the percentage mean ± SD (* *p* < 0.05, *** *p* < 0.001, **** *p* < 0.0001).

**Figure 2 ijms-25-06185-f002:**
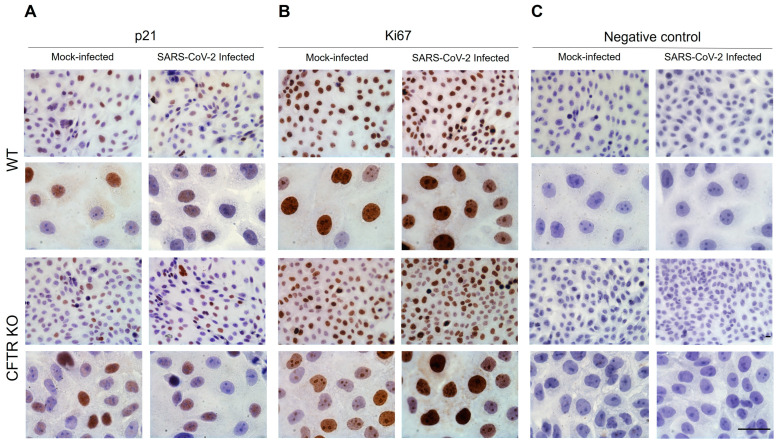
Immunohistochemical analysis of cellular senescence hallmarks in mock- and SARS-CoV-2-infected WT and CFTR KO 16HBE14o-cells. (**A**) p21 immunoreactivity in mock- and SARS-CoV-2-infected WT and CFTR KO 16HBE14o-cells after 48 h of culture. (**B**) Ki67 immunoreactivity in mock- and SARS-CoV-2-infected WT and CFTR KO 16HBE14o-cells after 72 h of culture. (**C**) Negative control in mock- and SARS-CoV-2-infected WT and CFTR KO cells at 72 h of culture. Bars: 15 µm.

**Figure 3 ijms-25-06185-f003:**
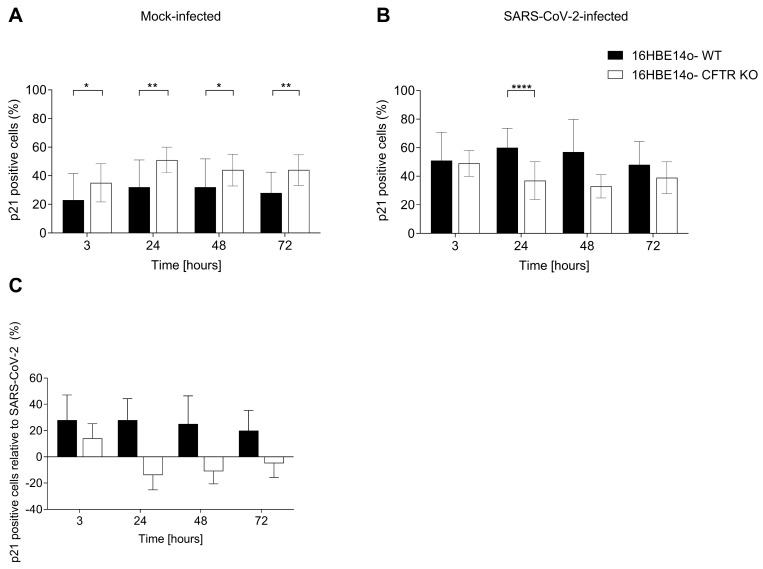
Percentage of cells expressing p21 in mock- and SARS-CoV-2-infected 16HBE14o-cell lines. (**A**) Percentage of p21-positive cells in mock-infected WT and CFTR KO 16HBE14o-cells. (**B**) Percentage of p21-positive cells in SARS-CoV-2-infected WT and CFTR KO 16HBE14o-cells. (**C**) Specific SARS-CoV-2 impact on p21-positive cells, obtained by subtracting the percentage of p21-positive cells in mock infection conditions from that after SARS-CoV-2 infection. Data are presented as the percentage mean ± SD of at least 10 frames for each experimental condition (* *p* < 0.05, ** *p* < 0.01, **** *p* < 0.0001).

**Figure 4 ijms-25-06185-f004:**
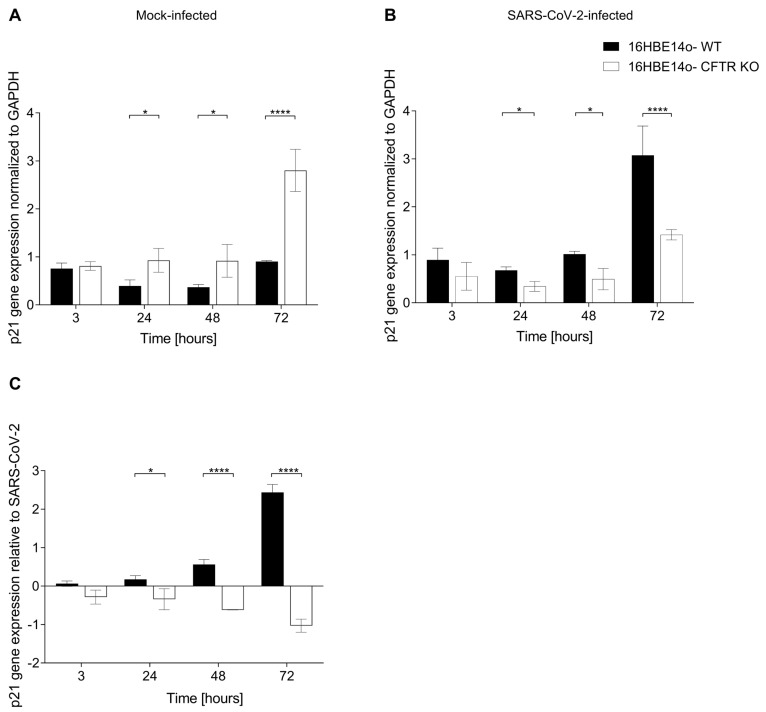
p21 gene expression in mock- and SARS-CoV-2-infected 16HBE14o-cells. (**A**) p21 gene expression in mock-infected WT and CFTR KO 16HBE14o-cells. (**B**) p21 gene expression in SARS-CoV-2-infected WT and CFTR KO 16HBE14o-cells. (**C**) Specific SARS-CoV-2 impact on p21 gene expression obtained by subtracting the normalized p21 gene expression in mock-infected cells from that recorded after SARS-CoV-2 infection. Gene expression was normalized to GAPDH (used as a housekeeping control), and data are presented as the percentage mean ± SD of at least three different experiments (n = 3; * *p* < 0.05, **** *p* < 0.0001).

**Figure 5 ijms-25-06185-f005:**
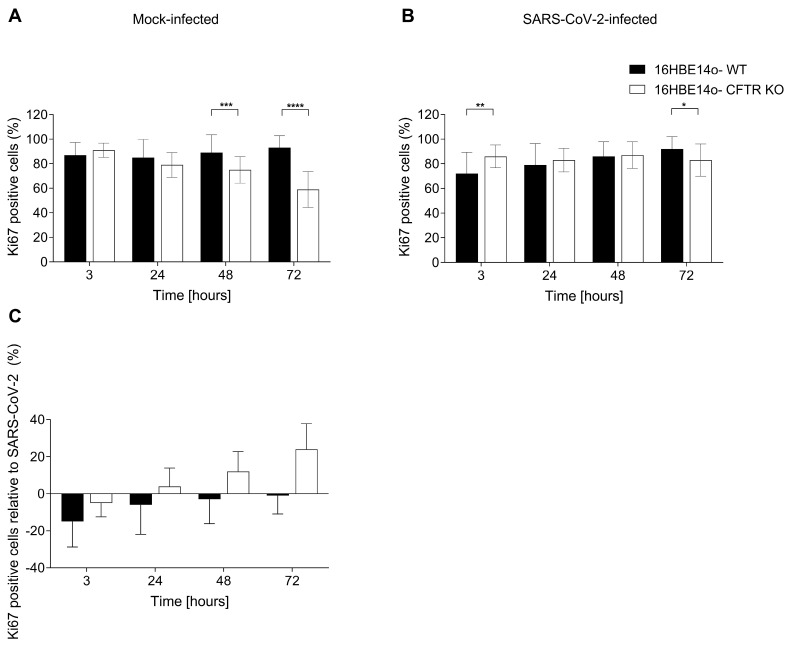
Percentage of cells expressing Ki67 in mock- and SARS-CoV-2-infected 16HBE14o-cell lines. (**A**) Percentage of Ki67-positive cells in mock-infected WT and CFTR KO 16HBE14o-cells. (**B**) Percentage of Ki67-positive cells in SARS-CoV-2-infected WT and CFTR KO 16HBE14o-cells. (**C**) Specific SARS-CoV-2 impact on Ki67^+^ cells obtained by subtracting the percentage of Ki67^+^ cells in the mock infection from that after SARS-CoV-2 infection. Data are presented as the percentage mean ± SD of at least 10 frames for each experimental condition (* *p* < 0.05, ** *p* < 0.01, *** *p* < 0.001, **** *p* < 0.0001).

**Figure 6 ijms-25-06185-f006:**
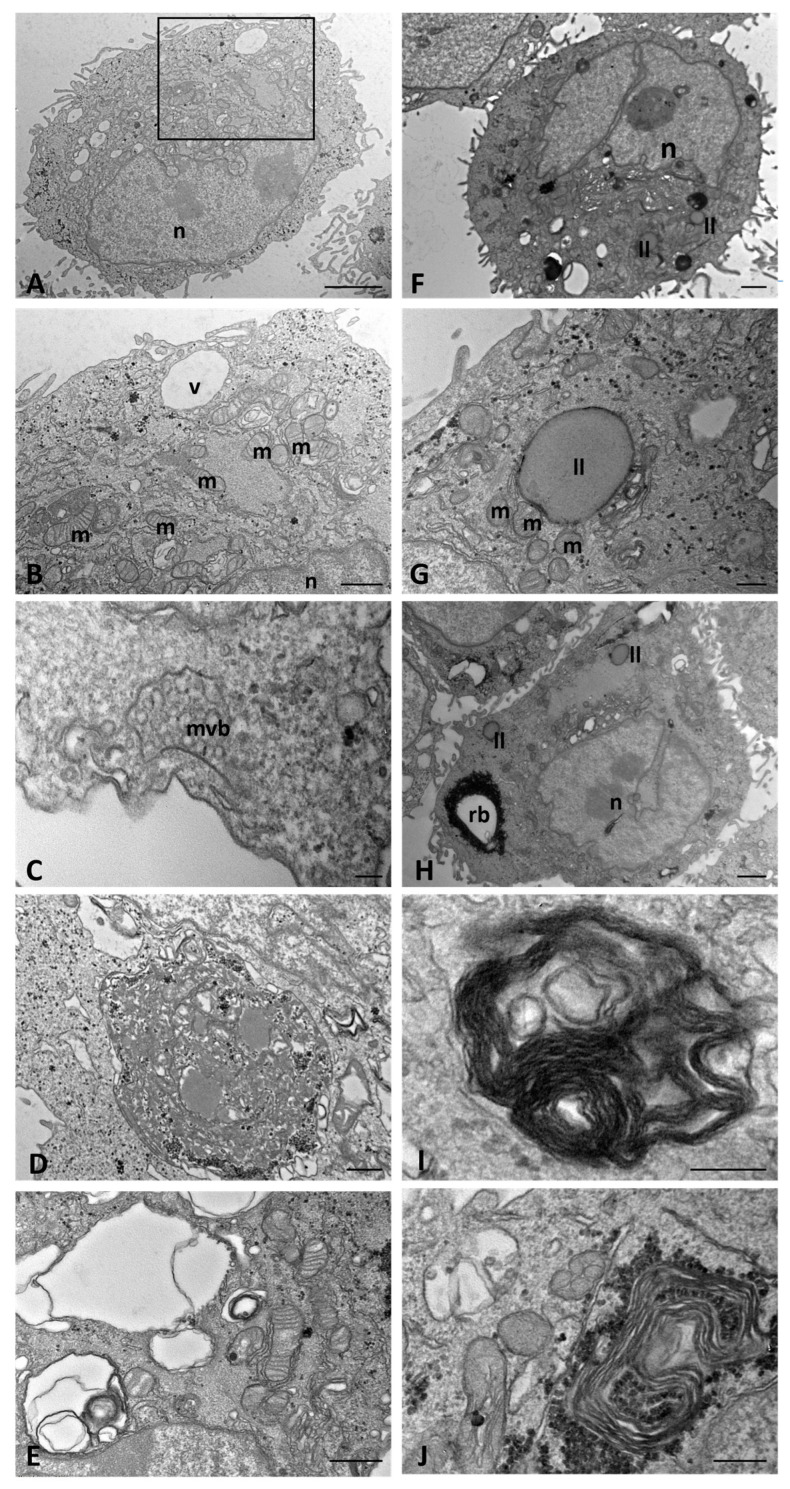
Transmission electron microscopy showing the morphology of mock-infected WT (**A**–**E**) and CFTR KO (**F**–**J**) 16HBE14o-cells at an early time of culture. ll(s): lipolysosome(s), m: mitochondrion, mvb: multivesicular body, n: nucleus, rb: residual body, v: vacuole. The boxed area in (**A**) is shown at a higher magnification in (**B**). Bars: (**A**,**F**,**H**) 1 µm; (**B**,**D**,**E**) 500 nm; (**G**,**I**,**J**) 200 nm; (**C**) 100 nm.

**Figure 7 ijms-25-06185-f007:**
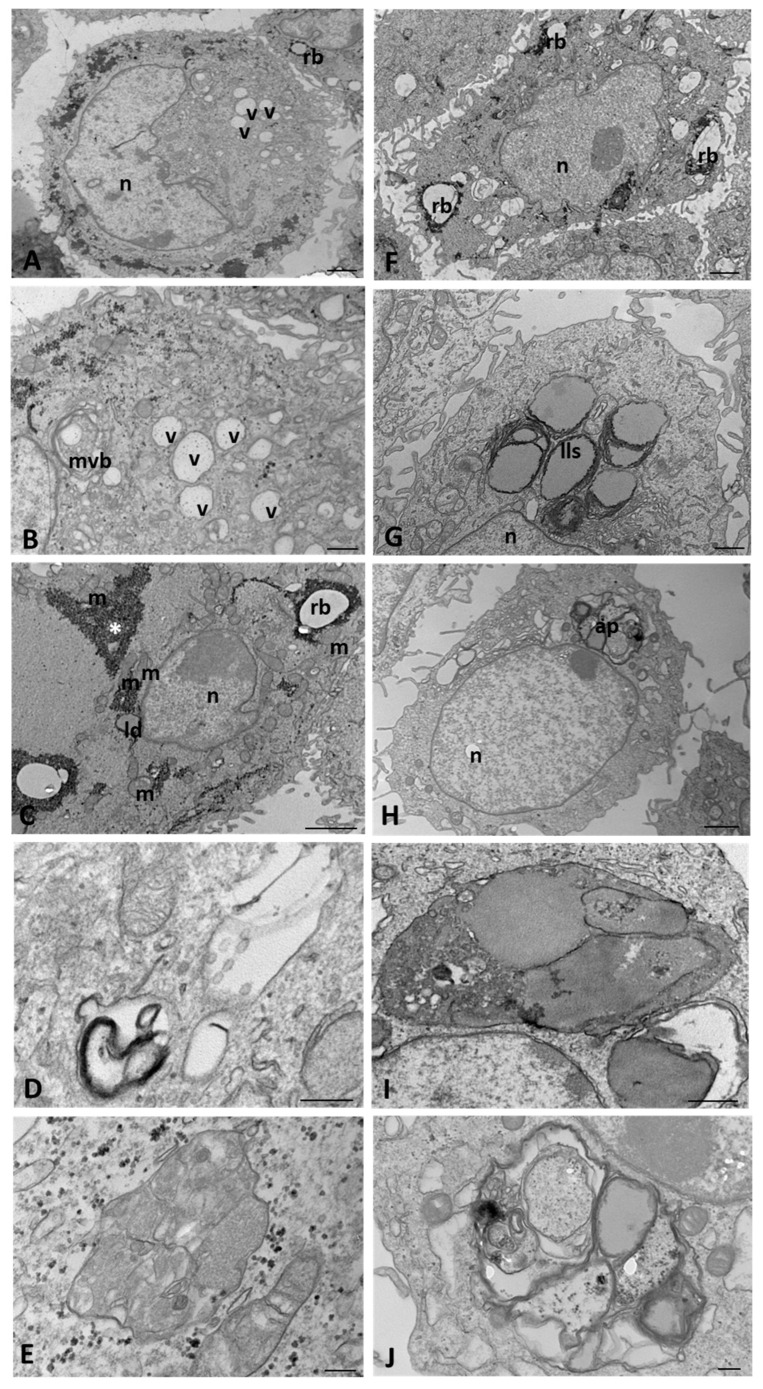
Transmission electron microscopy showing the morphology of mock-infected WT (**A**–**E**) and CFTR KO (**F**–**J**) 16HBE14o-cells at a late time of culture. ap: autophagosome, asterisk: electron-dense aggregate, ld: lipid droplet, ll(s): lipolysosome(s), m: mitochondrion, mvb: multivesicular body, n: nucleus, rb: residual body, v: vacuole Bars: (**A**,**C**,**F**,**H**) 1 µm; (**B**,**G**,**I**) 500 nm; (**D**,**E**,**J**) 200 nm.

**Figure 8 ijms-25-06185-f008:**
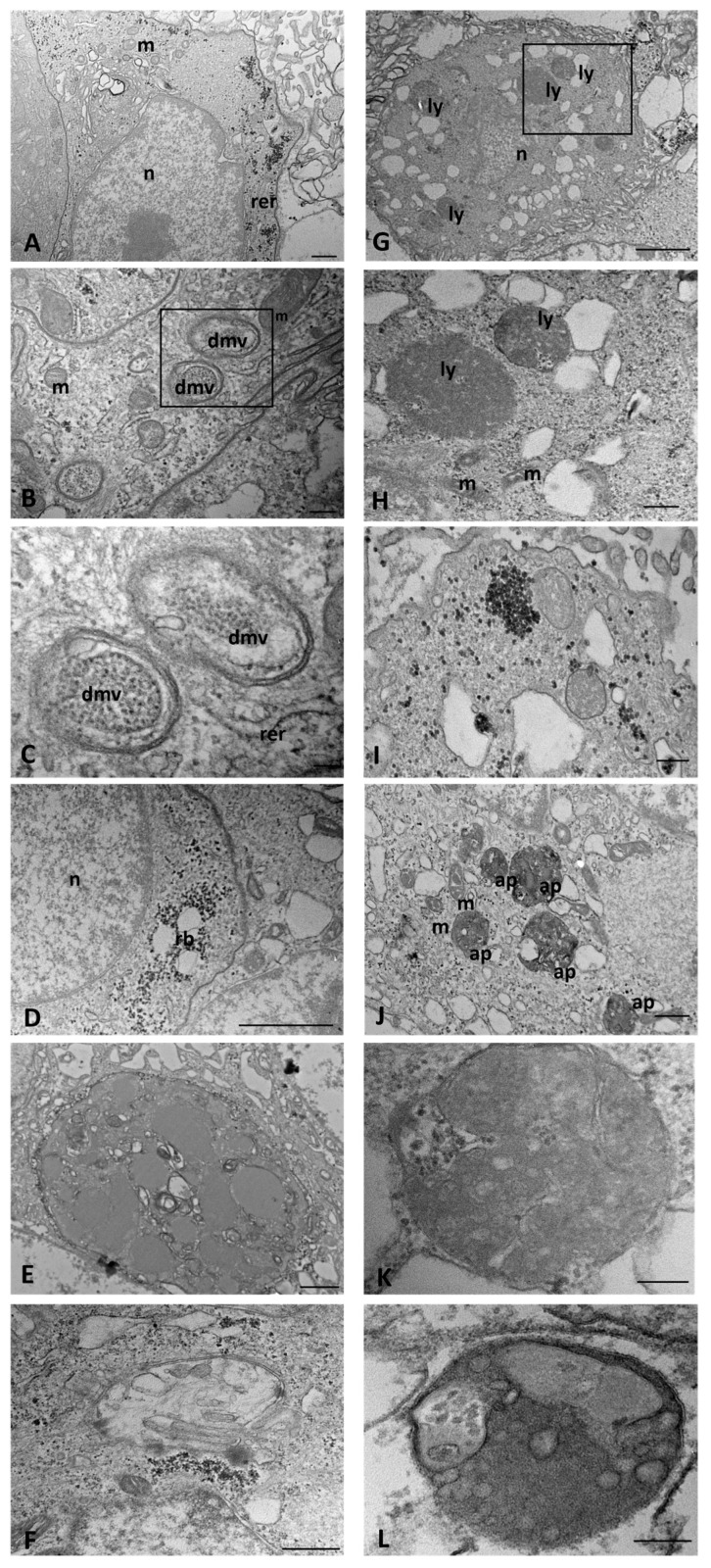
Transmission electron microscopy showing the morphology of SARS-CoV-2-infected WT (**A**–**F**) and CFTR KO (**G**–**L**) 16HBE14o-cells at an early stage of infection. ap: autophagosome, dmv: double-membrane vesicle, ly: lysosome, m: mitochondrion, n: nucleus, rb: residual body, rer: rough endoplasmic reticulum. The boxed areas in (**B**,**G**) are shown at higher magnification in (**C**,**H**) respectively. Bars: (**E**) 1 µm; (**A**,**D**,**F**–**H**,**J**) 500 nm; (**B**) 200 nm; (**C**,**I**,**K**,**L**) 100 nm.

**Figure 9 ijms-25-06185-f009:**
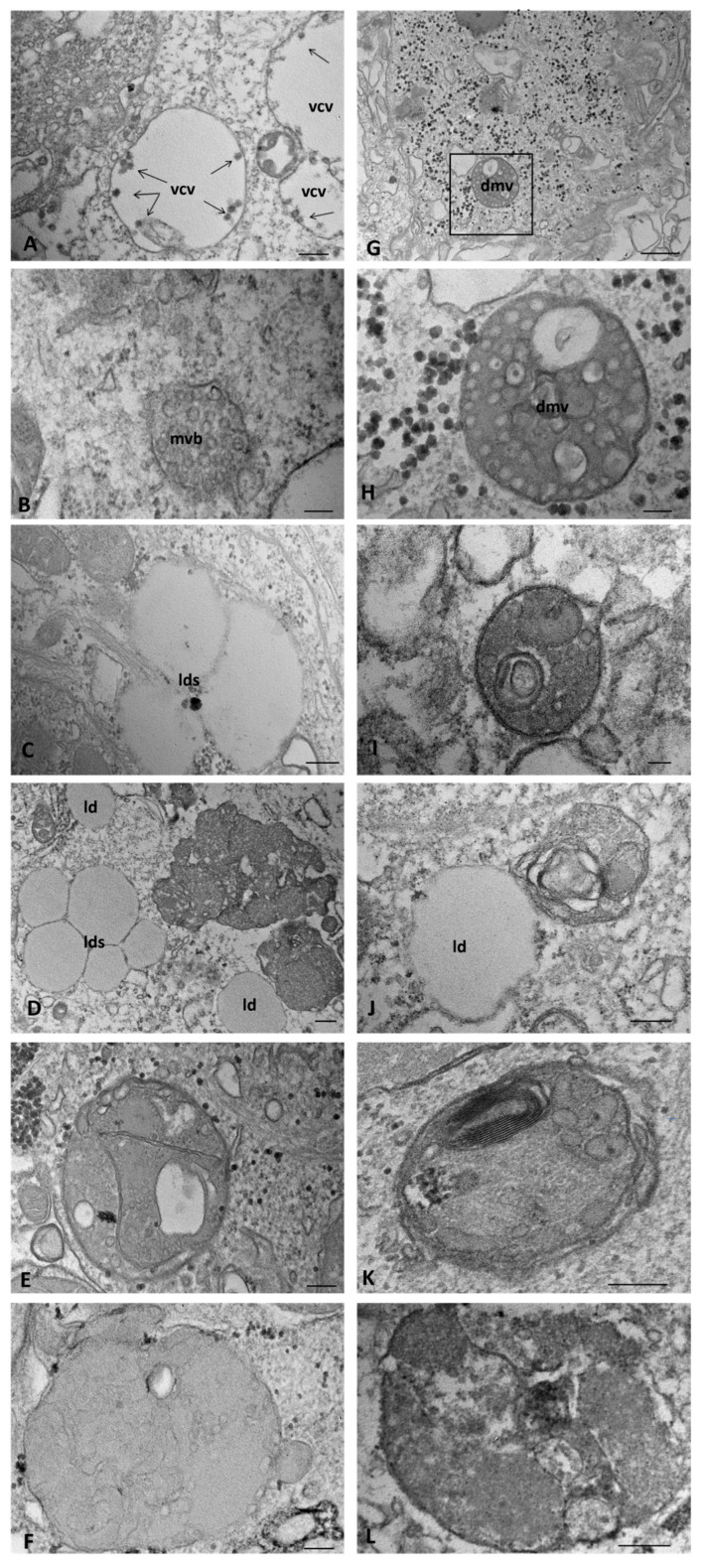
Transmission electron microscopy showing the morphology of SARS-CoV-2-infected WT (**A**–**F**) and CFTR KO (**G**–**L**) 16HBE14o-cells at a late stage of infection. dmv: double-membrane vesicle, ld(s): lipid droplet(s), mvb: multivesicular body, vcv: viron-containing vesicle. The boxed area in (**G**) is shown at higher magnification in (**H**). Bars: (**G**) 500 nm; (**A**,**C**,**D**,**J**,**L**) 200 nm; (**B**,**E**,**F**,**H**,**K**) 100 nm; (**I**) 50 nm.

**Figure 10 ijms-25-06185-f010:**
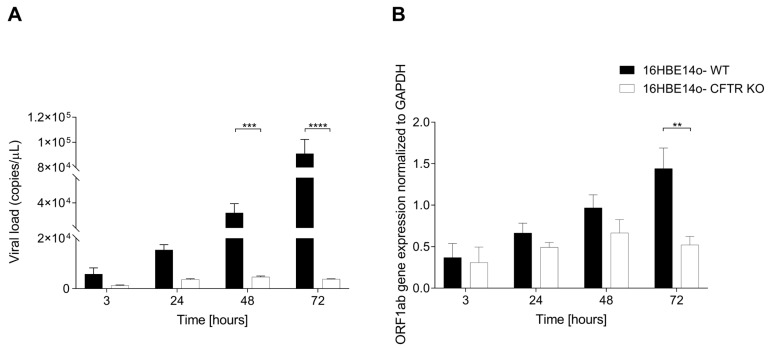
Extra- and intracellular analysis of the SARS-CoV-2 content. (**A**) Supernatant measurement of SARS-CoV-2 viral load using commercial qualitative real-time RT–PCR. (**B**) Intracellular measurement of ORF-1ab mRNA expression normalized to GAPDH. Data represent mean ± SD from n = 3 independent experiments (** *p* < 0.01, *** *p* < 0.001, **** *p* < 0.0001).

**Table 1 ijms-25-06185-t001:** Description of the main morphological characteristics in mock-infected WT and CFTR-KO 16HBE14o-cells at the early and late stages of analysis.

Time of Analysis	16HBE14o- Cells	Lipid Droplets	Lipolysosomes	Autophagosomes
Early	WT	**●**	**-**	**-**
	CFTR KO	**●**	**●**	**-**
Late	WT	**-**	**-**	**●**
	CFTR KO	**●**	**●**	**●**

Table legend: **●**: present; **-**: not present.

**Table 2 ijms-25-06185-t002:** Description of main morphological characteristics in SARS-CoV-2-infected WT and CFTR-KO 16HBE14o-cells at the early and late stages of analysis.

Time of Analysis	16HBE14o- Cells	DMVs	Virions	Lipid Droplets	Lipolysosomes	Autophagosomes	
						Cellular Material	Replicative Structure
Early	WT	**●**	**-**	**-**	**-**	**●**	**-**
	CFTR KO	**●**	**-**	**●**	**-**	**●**	**-**
Late	WT	**●**	**●**	**●**	**-**	**●**	**●**
	CFTR KO	**●**	**-**	**●**	**-**	**●**	**-**

Table legend: **●**: present; **-**: not present.

## Data Availability

The data presented in this study are available on request from the corresponding author.

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
