# Peer review of "Loss of CFTR Reverses Senescence Hallmarks in SARS-CoV-2 Infected Bronchial Epithelial Cells"

_ijms, 2024, doi:10.3390/ijms25116185_

Round 1

Reviewer 1 Report

Comments and Suggestions for Authors

Merigo et al. observed in their study that CFTR knockout exacerbates senescence in uninfected cells, while SARS-CoV-2 infection activates senescence in wild-type cells but appears to suppress it in CFTR-deficient cells. Althoguh these findings are intriguing, several key experimental aspects require further clarification.

1. CFTR was genetically knocked out in 16HBE14o- cells. It is essential to provide a concise overview of CFTR's physiological role and its implications in cellular senescence.

2. Although CFTR knockout cells were generated in this investigation, there is a notable absence of data confirming the successful CFTR knockout. Evaluation of CFTR protein expression in both wild-type and CFTR knockout cells is imperative to validate the knockout model.

3. Notably, in CFTR-knockout cells, there was a significant reduction in the viral load of SARS-CoV-2. It is crucial to elucidate which specific step(s) of the viral life cycle are affected by CFTR knockout. If inhibition occurs before viral entry, resulting in substantial population of uninfected cells post-inoculation, the impact of SARS-CoV-2 infection on senescence could be minimized. Therefore, investigating the specific viral life cycle stages affected by CFTR knockout is essential for a comprehensive understanding of the obstarved phenomena.

Author Response

Dear Reviewer,

We sincerely thank You for the time and dedication to this manuscript. The suggestions and constructive comments have improved the report and deepened the impact of the findings. Point-by-point responses to reviewers’ comments are reported below:

Merigo et al. observed in their study that CFTR knockout exacerbates senescence in uninfected cells, while SARS-CoV-2 infection activates senescence in wild-type cells but appears to suppress it in CFTR-deficient cells. Althoguh these findings are intriguing, several key experimental aspects require further clarification.

  1. CFTR was genetically knocked out in 16HBE14o- cells. It is essential to provide a concise overview of CFTR's physiological role and its implications in cellular senescence.

Thank you for the suggestion. Lines 58-66.

  1. Although CFTR knockout cells were generated in this investigation, there is a notable absence of data confirming the successful CFTR knockout. Evaluation of CFTR protein expression in both wild-type and CFTR knockout cells is imperative to validate the knockout model.

Many thanks for raising this important issue.

The cell lines used in our manuscript were provided by Dr Roberto Plebani and Prof Mario Romano, co-authors of the manuscript.

The CRISP-Cas 9 based CFTR KO procedure was first reported by Dr Plebani in an abstract presented at the North American Cystic Fibrosis Conference 2020 (reference 70). The full paper describing in detail the procedure is under submission.

We have now added more information in the materials and methods section (lines 463-469) and a western blot of CFTR protein expression by the two cell lines in the supplementary materials (Supplementary Figure 4).

  1. Notably, in CFTR-knockout cells, there was a significant reduction in the viral load of SARS-CoV-2. It is crucial to elucidate which specific step(s) of the viral life cycle are affected by CFTR knockout. If inhibition occurs before viral entry, resulting in substantial population of uninfected cells post-inoculation, the impact of SARS-CoV-2 infection on senescence could be minimized. Therefore, investigating the specific viral life cycle stages affected by CFTR knockout is essential for a comprehensive understanding of the obstarved phenomena.

This is an important, albeit quite complex, issue.  We provide evidence that the viral cycle is delayed even if, we agree with this reviewer, we cannot precisely define at which step(s). In this study we intended to explore the senescent phenotype and whether there was an influence of the virus on it (indeed it was). It is more difficult prove that the senescent phenotype is responsible for the reduced capability of CFTR-KO cells to produce viral particles. Notwithstanding, this was not the main focus of this manuscript, nevertheless, trying to answer the reviewer’s question, we have added some additional observations, namely flow cytometry analysis of ACE2 basal membrane expression and IHC analysis of SARS-CoV-2 nucleocapsid protein, that might better define the described phenomenon. ACE2 protein expression was higher in WT compared to CFTR KO cells (Supplementary Figure 2). The IHC analysis of nucleocapsid protein led us to conclude that both WT and CFTR KO 16HBE14o- cells are permissive to SARS-CoV-2, since we observed a reactivity to SARS-CoV-2 Nucleocapsid antibody at all infection time points analyzed, even though the reactivity seemed to be more marked in WT than in KO cells (Supplementary Figure 3). This observation reflects the reduced viral production detected in the cell culture supernatants. Altogether, we hypothesize that multiple mechanisms may contribute to the reduced virion production and pathogenicity in CF cells, among which 1) reduced expression of ACE2 that can limit the viral entry; 2) the presence of a less permissive intracellular environment, associated to a senescent phenotype, as demonstrated by morphological data.

Please, find the results described in results section 2.6. and the relative figures reported in the supplementary materials section (Supplementary Figure 2 and 3).  We also discussed the results at lines 417-428.

Reviewer 2 Report

Comments and Suggestions for Authors

The current study presents intriguing findings utilizing CFTR KO cells. However, there are certain areas of concern regarding the quantitative analysis and sample size indication, which require attention. Below are specific points that need to be addressed:

Figures 2 and 3: It is unclear how many cells were analyzed for quantification. Additionally, the sample size is not clearly indicated.

Figure 3: Error bars should be included.

Figure 3: Quantification of positive cells at various time points is mentioned, but the authors should provide raw data (immunostaining data) for clarity in the quantification process.

Figure 4A: The authors should provide an explanation for the increase in p21 expression observed after 72 hours in mock-infected cells. Additionally, this result should address the inconsistency noted in Figure 3A, where p21 levels remain unchanged.

Tables 1 and 2: The criteria defining "detectable," "present," and "abundant" are unclear. It's essential to analyze a sufficient number of cells for quantitative analysis to ensure robustness of the findings.

Author Response

Dear Reviewer,

We sincerely thank You  for the time and dedication to this manuscript. The suggestions and constructive comments have improved the report and deepened the impact of the findings. Point-by-point responses to reviewers’ comments are reported below:

The current study presents intriguing findings utilizing CFTR KO cells. However, there are certain areas of concern regarding the quantitative analysis and sample size indication, which require attention. Below are specific points that need to be addressed:

Figures 2 and 3: It is unclear how many cells were analyzed for quantification. Additionally, the sample size is not clearly indicated.

Many thanks for the mentioning this point. The number of cells analyzed are reported in the Supplementary table 1 (p21) and 2 (ki67).

Sample size is two cell lines with a minimum of three biological replicates.

Figure 3: Error bars should be included.

Thanks for the suggestion. We have added the error bars in Figures 3 and 5.

Figure 3: Quantification of positive cells at various time points is mentioned, but the authors should provide raw data (immunostaining data) for clarity in the quantification process.

Many thanks for raising this point. The raw data are reported in supplementary materials, Supplementary Table 1 and 2.

Figure 4A: The authors should provide an explanation for the increase in p21 expression observed after 72 hours in mock-infected cells. Additionally, this result should address the inconsistency noted in Figure 3A, where p21 levels remain unchanged.

As we reported both in the introduction and in the discussion, there are some previously published papers (ref. 26 and 56) reporting an increased expression of senescence markers in CFTR modified cells. Moreover, the apparent inconsistency between Figures 3A and 4A is probably due to the different type of analysis: in Figure 3A, we represented the number of cells expressing the p21 protein, whereas in Figure 4A, we measure gene expression by quantifying the relative mRNA found in the cell lysate. As not all mRNA is translated into protein a discrepancy in the two analyses is likely to occur.

Tables 1 and 2: The criteria defining "detectable," "present," and "abundant" are unclear. It's essential to analyze a sufficient number of cells for quantitative analysis to ensure robustness of the findings.

Thank you very much for this suggestion. Our ultrastructural analysis is indeed qualitative, not quantitative; therefore, it is more correct to use the terminology presence/absence of the various structures that has a strong influence on cell behavior and features. We have therefore modified tables 1 and 2 accordingly.

Round 2

Reviewer 1 Report

Comments and Suggestions for Authors

Supplementary Figure 3 indicates that the viral N protein is expressed in almost all cells inoculated with SARS-CoV-2. This result suggests that CFTR KO does not affect viral entry, a conclusion I fully agree with, supporting the idea that SARS-CoV-2 inhibits senescence. However, Supplementary Figure 2 demonstrates that CFTR KO induces a reduction in the expression of the infection receptor ACE2, suggesting that viral entry may be diminished. These results are inconsistent and confound the conclusion. Therefore, I think that Supplementary Figure 2 is not necessary for this study.

Reviewer 2 Report

Comments and Suggestions for Authors

The revised manuscript has been substantially improved. 

All my queries are appropriately addressed.